# LEOPARD🐆: A VISION LANGUAGE MODEL FOR TEXT-RICH MULTI-IMAGE TASKS

## ABSTRACT

Text-rich images, where text serves as the central visual element guiding the overall understanding, are prevalent in real-world applications, such as presentation slides, scanned documents, and webpage snapshots. Tasks involving multiple text-rich images are especially challenging, as they require not only understanding the content of individual images but reasoning about inter-relationships and logical flows across multiple visual inputs. Despite the importance of these scenarios, current multimodal large language models (MLLMs) struggle to handle such tasks due to two key challenges: (1) the scarcity of high-quality instruction tuning datasets for text-rich multi-image scenarios, and (2) the difficulty in balancing image resolution with visual feature sequence length. Low-resolution encoding impairs the recognition of embedded text, while high-resolution encoding quickly exceeds the model's maximum sequence length under multi-image settings. To address these challenges, we propose LEOPARD, a MLLM designed specifically for handling vision-language tasks involving multiple text-rich images. First, we curated about one million high-quality multimodal instruction-tuning data, tailored to text-rich, multi-image scenarios. Second, we developed an adaptive high-resolution multi-image encoding module to dynamically optimize the allocation of visual sequence length based on the original aspect ratios and resolutions of the input images. Experiments across a wide range of benchmarks demonstrate our model's superior capabilities in text-rich, multi-image evaluations and competitive performance in general domain evaluations. We are committed to open-source models and will release all collected data, code, and checkpoints to the community[1].

## 1 INTRODUCTION

Multimodal large language models (MLLMs) have revolutionized vision-language tasks, driving advancements in a variety of areas such as image captioning and object detection (Wang et al., 2023b; Zhang et al., 2024; Zang et al., 2024). These improvements extend to applications involving *text-rich images* where text serves as the primary visual element guiding image comprehension, such as visual document understanding (Mathew et al., 2021) and scene text recognition (Singh et al., 2019b). Traditional OCR-based pipelines in these text-rich visual scenarios are being replaced by end-to-end approaches that directly encode intertwined multimodal inputs (Wu et al., 2023b; Zhang et al., 2023; Tang et al., 2024), leading to improved efficiency and accuracy in handling text-rich images.

Despite these advancements, the majority of existing open-source MLLMs, like LLaVAR (Zhang et al., 2023) and mPlug-DocOwl-1.5 (Hu et al., 2024a), have primarily focused on optimizing performance for text-rich *single-image tasks*. This focus inherently limits their applicability in many real-world scenarios, where tasks often involve *multiple inter-connected images*. For instance, multi-page visual document understanding requires integrating information spread across different pages to capture the logical flow across the whole document (Tito et al., 2022; Landeghem et al., 2023). To understand presentation slides, grasping the overarching narrative necessitates understanding multiple slides with unique but interrelated content (Tanaka et al., 2023). These vision-language tasks on multiple text-rich images require advanced capabilities that go beyond merely recognizing text and visuals within a single image; they involve understanding and reasoning about relationships and

---

[1] https://anonymous.4open.science/r/Leopard-8E26/.

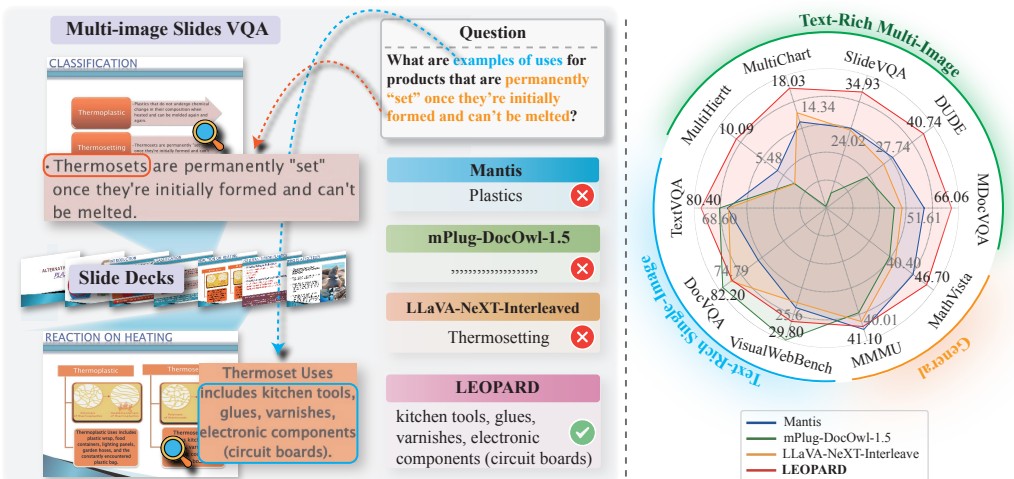

Figure 1: Left: A demonstration of a text-rich multi-image task. Models need to reason about the textual content across multiple images to answer the question correctly. LEOPARD successfully generates the right answer while baselines fail. Right: Evaluation results of LEOPARD and three baselines. Our model surpasses its counterparts across text-rich multi-image benchmarks by a large margin, maintaining comparable performance on single and general evaluations.

logical flows across multiple visual inputs. While some models – such as OpenFlamingo (Awadalla et al., 2023), VILA (Lin et al., 2023), Idefics2 (Laurençon et al., 2024b) – have made strides toward supporting multi-image inputs, they mainly focus on scenarios with natural images but fall short in understanding sequences of *text-rich images* with interrelated textual and visual information. We plot the performance of representatives of the aforementioned models in Figure 1. Upon examining their training data and model architecture, we identified two primary limitations within these models.

First, there is a scarcity of high-quality instruction tuning datasets on text-rich multi-image scenarios. Existing visual instruction tuning datasets for text-rich images are predominantly based on single-image inputs (Kafle et al., 2018; Singh et al., 2019b; Masry et al., 2022; Tang et al., 2024), which limits the model ability to generalize and reason across multiple images. Second, in text-rich multi-image scenarios, there is a challenge of balancing image resolution and sequence length limitations. Many general-domain MLLMs adopt the low-resolution settings of pre-trained visual encoders (Lin et al., 2023; Jiang et al., 2024). However, for text-rich images, such as scientific reports, recognizing text content becomes difficult at low resolutions. While some approaches overcome this in single-image settings by splitting the original image to preserve high-resolution details (Liu et al., 2024a; Hu et al., 2024a), this approach is less effective when applied to multiple images, as it quickly exceeds model's maximum sequence length. Moreover, compressing such long-sequence representations into shorter ones leads to significant information loss, thereby degrading model performance (Awadalla et al., 2023; Laurençon et al., 2023). Thus, a critical balance must be struck between maintaining sufficient visual detail and keeping sequence lengths manageable.

In this paper, we introduce a novel multimodal large language model, named **LEOPARD**[2]. LEOPARD is specifically designed to handle complex *text-rich, multi-image* tasks. To train LEOPARD, we first curated **about one million high-quality multimodal instruction-tuning data**, tailored to the text-rich, multi-image scenarios. This dataset spans three key domains that are commonly encountered in real-world scenarios: (1) multi-page documents, (2) multi-charts and multi-tables, (3) webpage trajectories. These scenarios capture the increasing complexity and multimodal nature of modern digital information. In addition, to enable high-resolution encoding in multi-image inputs, we equipped LEOPARD with an **adaptive high-resolution multi-image encoding module**. Specifically, it dynamically optimizes the allocation of visual sequence length based on the original aspect ratios and resolutions of the input images. We then apply pixel shuffling to losslessly compress (Chen

---

[2]Leopards have remarkable visual adaptations that allow them to track prey both from afar and up close, making them highly efficient hunters.

et al., 2024a) long visual feature sequences into shorter ones. This approach allows the model to accommodate multiple high-resolution images without compromising detail or clarity.

We conducted experiments on 13 vision-language benchmark datasets, evaluating LEOPARD from multiple perspectives. Consistent improvements were observed when training LEOPARD with two distinct base model architectures: LLaVA and Idefics2. Our results demonstrate LEOPARD's superior performance on 5 text-rich, multi-image benchmarks, outperforming the best open-source MLLM by an average of **+9.61** points. Moreover, LEOPARD remains highly competitive in text-rich single-image tasks and general-domain vision-language benchmarks, achieving comparable results to state-of-the-art MLLMs without extensive fine-tuning. Further ablation studies confirm the effectiveness of our instruction-tuning dataset and the adaptive high-resolution encoding module. These findings highlight LEOPARD's strong performance and versatility across various multimodal applications.

## 2 RELATED WORK

**Multimodal Large Language Models (MLLMs).** Many approaches have been proposed for building MLLMs, leveraging different architectural designs. A widely adopted approach is the decoder-only architecture, exemplified by LLaVA (Liu et al., 2023b), Emu2 (Sun et al., 2023), and Intern-VL (Chen et al., 2024b). These models typically incorporated a visual encoder to encode images, a vision-language connector to project visual features into the language feature space, and a language model that processes both visual and textual information jointly. Another line of work employed cross-attention architectures where encoded image features are integrated with textual tokens via cross-attention layers, as seen in Flamingo (Alayrac et al., 2022), OpenFlamingo (Awadalla et al., 2023) and CogVLM (Wang et al., 2023a). Such a design allows models to retain the benefits of a fully intact language model but introduces new parameters to manage the visual-textual interplay.

**Text-rich MLLMs.** Text-rich images are traditionally processed in pipelines (Singh et al., 2019a; Hu et al., 2020), where an OCR module first recognized text from the image, followed by processing through a language model. To improve efficiency and avoid error propagation, with the advent of MLLMs, end-to-end approaches become more popular recently. For instance, LLaVAR (Zhang et al., 2023) utilized a dataset of 400K instances with OCR-enhanced text to outperform LLaVA on various text-rich VQA tasks. Subsequent models such as UReader (Ye et al., 2023), TextMonkey (Liu et al., 2024d), and Mplug-DocOwl-1.5 (Hu et al., 2024a) recognized the importance of high-resolution encoding for accurate text comprehension, so they adopted strategies that cropped single images into multiple sub-images to preserve the original resolution during visual encoding. However, these approaches are primarily trained on single-image data, and struggle to generalize effectively to multi-image scenarios. Furthermore, the straightforward partitioning technique encounters challenges with multi-image inputs, as the sequence length rapidly increases with the number of images.

**Multi-image MLLMs.** Efforts have been made in training MLLMs with multi-image inputs due to the prevalence of multi-image scenarios in real-world applications. Mantis (Jiang et al., 2024) introduced a multi-image instruction tuning dataset on a variety of natural image scenarios. Besides, both VILA (Lin et al., 2023) and Idefics-2 (Laurençon et al., 2024b) incorporated image-text interleaved data during their pre-training. LLaVA-Next-Interleave (Li et al., 2024c) further extended this by incorporating videos and multi-view 3D data into the training pipeline. However, these works primarily target natural images and general visual understanding, leaving a gap in handling text-rich, multi-image scenarios. Natural images typically follow a different distribution from text-rich images and often do not demand high-resolution processing. As a result, many existing multi-image MLLMs struggle to generalize to text-rich scenarios. Our work aims to address this gap by specifically focusing on multi-image settings where text-rich images are the primary input.

**Concurrent Works Released in 08/2024 and 09/2024.** Very recently, multi-image training for MLLMs has attracted intense attention from researchers. Several concurrent efforts have included multi-image interleaved data to train their models, such as LLaVA-OneVision 08/2024 (Li et al., 2024b), Idefics3 (08/2024, Laurençon et al., 2024a), NVLM (09/2024, Dai et al., 2024), mPlug-DocOwl-2 (09/2024, Hu et al., 2024b), Molmo (09/2024, Deitke et al., 2024) and Qwen2-VL (09/2024, Wang et al., 2024). This trending paradigm highlights the significant practical value of multi-image MLLMs by enhancing their ability to tackle a wide range of real-world applications. The incorporation of multi-image instruction tuning data is therefore of paramount importance.

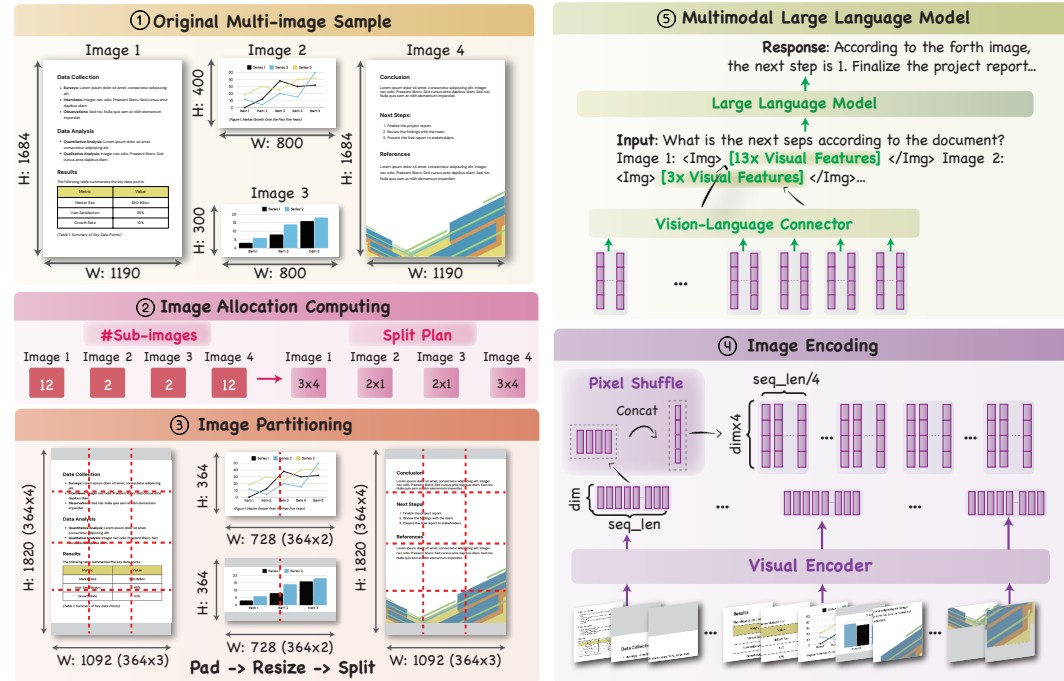

Figure 2: The overall model pipeline. Given ① raw image inputs, ② we first compute the optimal allocation of sub-image numbers and splitting strategy for all images based on their resolution and aspect ratio. ③ The images undergo padding, resizing, and splitting operations. ④ Both sub-images and resized original images are then encoded into a sequence of visual features. These sequences subsequently undergo a pixel shuffle operation that concatenates every four features. ⑤ The visual features are projected into the language embedding space via a vision-language connector. Finally, the large language model then integrates these visual and language embeddings to generate responses.

## 3 METHOD

LEOPARD follows the typical design of decoder-only vision language models (Liu et al., 2023b; 2024a; Li et al., 2024c), including a visual encoder, a vision language connector, and a language model (LM), as shown in Figure 2 (④⑤). Specially, the input images are first passed through the visual encoder, which extracts high-level visual features and captures essential semantic information. These visual features are then projected into the language representation space via the vision-language connector. After this transformation, the visual tokens are interleaved with the textual tokens, resulting in a sequence of interleaved text-visual tokens. This interleaved sequence is then fed into the LM, which processes these inputs in a causal manner, leveraging the contextual dependencies between text and visual information to generate coherent outputs that align with both modalities.

### 3.1 MULTI-IMAGE TEXT-RICH INSTRUCTION TURNING DATASET

To train LEOPARD, we construct a large instruction-tuning dataset named LEOPARD-INSTRUCT, comprising **925K** instances, with **739K** specifically designed for text-rich, multi-image scenarios. While we extensively surveyed existing open-source datasets, we only identified **154K** usable text-rich, multi-image samples, which is far from sufficient for effective instruction tuning, as shown in prior MLLM studies (Jiang et al., 2024; Laurençon et al., 2024b; Li et al., 2024c). To address this data scarcity, we developed several data collection pipelines to collect high-quality text-rich, multi-image data, resulting in additional **585K** instances. Each instance consists of a set of images along with corresponding task instructions and responses. The dataset details are presented in Table 1, and a detailed breakdown of its composition can be found in Appendix A.1.

**Documents and Slides** are common sources of multi-image data that primarily contain text and require cross-page context integration to fully understand the information.

These data is collected in three ways. First, we include 69K public multi-page document and slide datasets (Tito et al., 2022; Landeghem et al., 2023; Zhu et al., 2022; Tanaka et al., 2023), covering a variety of document types such as scanned handwriting, printed documents, and digital PDFs. Second, we adapt two single-page document datasets, DocVQA (Mathew et al., 2021) and ArxivQA (Li et al., 2024d), for multi-image settings. Following Jiang et al. (2024), we randomly merge 2 to 4 single-page instances by concatenating their respective images and Q-A pairs. Prompts like "in the second image" are added to direct the model's focus to the appropriate image. These merged samples help the model learn how natural language references align with corresponding image features. Third, we collect raw slides from Sefid et al. (2021) and SlideShare[3], and use GPT-4o to generate Q-A pairs and reasoning steps. We show the prompt to GPT in Figure 5. Upon manually reviewing 100 instances annotated by GPT-4o, we found an accuracy rate over 90%, indicating high annotation quality.

**Tables and Charts** provide highly organized, structured quantitative information, often involving complex data patterns and relationships, requiring the integration of both visual and textual elements for accurate interpretation.

To address the lack of instruction tuning data involving multiple tables or charts, we use the following strategies. First, we include 21K open-source multi-chart and multi-table datasets (Zhao et al., 2022; Pal et al., 2023), originally stored in JSON or DataFrame formats. We programmatically render these tables as images, converting them into multimodal data. Details of rendering can be found in Appendix A.3 Second, We utilize the TableGPT (Li et al., 2024e) dataset and split each table into multiple sub-tables, then convert them into figures, thereby creating multi-modal, multi-table instruction data. Third, we apply the same merging strategy used for combining single-page documents to synthesize multi-image datasets. This approach integrates several single-chart datasets, including ChartGemma (Masry et al., 2024), ChartQA (Masry et al., 2022), DVQA (Kafle et al., 2018), and FigureQA (Kahou et al., 2018). Besides, we generate new multi-chart data from social reports of the Pew Research Center[4] that feature multiple interrelated charts within the articles under the same topic. We download charts from the website and use GPT-4o to create 20K Q-A pairs that require multi-chart understanding.

Table 1: Data statistics of the LEOPARD-INSTRUCT dataset.

| Data Types | # Instances |
| --- | --- |
| **Total Samples** | 925K |
| Single-image | 186K (20.10%) |
| Multi-image | 739K (79.89%) |
| *Public | 154K (16.65%) |
| *New (Ours) | 585K (63.24%) |
| **Rationales** | |
| *Existing | 214K (23.14%) |
| *New (Ours) | 250K (27.02%) |
| *None | 461K (49.84%) |
| **Domains** | |
| Documents | 192K (20.76%) |
| Slide Decks | 16K (1.73%) |
| Tables | 48K (5.19%) |
| Charts | 353K (38.16%) |
| Webpages | 55K (5.95%) |
| Others | 261K (28.22%) |

**Webpage Snapshots** consist of sequential images representing web pages, providing visual context for user interactions and task flows. Understanding webpage is a critical skill for MLLMs to evolve into fully autonomous web agents (Deng et al., 2023; He et al., 2024). To collect and standardize relevant data, we format several web-related multimodal datasets into a Q-A structure as follows:

1. *Web action prediction data*: We include Mind2Web (Deng et al., 2023) and OmniACT (Kapoor et al., 2024), where we divide long web snapshots into multiple sub-figures, and plot bounding boxes based on the coordinates of web elements. Then GPT-4o is used to convert the original action data into a Q-A format, where the task is to identify the correct element to interact with.
2. *Web-based classification data*: We incorporate WebScreenshots (Aydos, 2020), WebVision (Li et al., 2017), and WebUI (Wu et al., 2023a). We utilize the web snapshots in these datasets and employ GPT-4o to generate Q-A pairs on webpage understanding, including chain-of-thought reasoning steps. The prompting details are provided in Figure 6.

**Augmenting with Rationales.** In contrast to single-image tasks, multi-image scenarios typically require MLLMs to integrate information across multiple images, making cross-image reasoning difficult to train when only the final answer is provided (Zheng et al., 2023; Hu et al., 2023). To address this, we employ GPT-4o to generate chain-of-thought (CoT) rationales for inherently multi-image datasets (excluding those formed by merging single-image data) that lack CoT annotations. This results in 250K instances with GPT-annotated reasoning, with the prompt detailed in Figure 7.

---

[3] https://www.slideshare.net
[4] https://www.pewresearch.org

**Other Domains.** We also include datasets from various other domains such as maps (MapQA, Chang et al., 2022), infographics (InfographicVQA, Mathew et al., 2022), mathematical diagrams (MathV360K, Shi et al., 2024), and abstractive diagrams (IconQA, Lu et al., 2021). We also incorporate mixed-domain datasets for text-rich images, including LLaVAR (Zhang et al., 2023), Monkey, Li et al., 2024f, and mPlugDocReason (Hu et al., 2024a). We remove duplicate subsets from these mixed-domain datasets. Among these datasets, 64K samples consist of multi-image data, while the remaining are single-image samples. To preserve natural image understanding ability, we add 313K samples from ShareGPT4V (Chen et al., 2023), an instruction dataset for natural images.

## 3.2 Adaptive High-resolution Multi-image Encoding

Image resolution significantly influences the visual perception and understanding capabilities of MLLMs, particularly when processing text-rich images. Low-resolution images often cause printed text to become blurred or unreadable, resulting in misinterpretations, perception errors, and visual hallucinations. The visual resolution of most existing MLLMs is determined by their pre-trained visual encoders, which are typically limited to low resolutions such as $224 \times 224$ or $336 \times 336$ pixels (Liu et al., 2023a; Lin et al., 2023; Jiang et al., 2024). These low-resolution constraints can hinder MLLMs to accurately understand textual information embedded within images.

To overcome these limitations, a natural solution is dividing a high-resolution image into multiple smaller sub-images, each of which is independently processed by the model's visual encoder (Liu et al., 2024a; Dong et al., 2024). This partitioning allows for the extraction of more fine-grained visual details, making it possible to capture small or densely packed textual elements. However, a major drawback of this approach is that it significantly increases the length of visual feature sequence. When applied to scenarios involving multiple image inputs, the feature sequences are easily exceeding the model's maximum sequence length limit. To address the issue, we follow the image-splitting idea and propose a novel adaptive high-resolution multi-image encoding strategy as follows.

**Image Allocation Computing**: To prevent the number of sub-image visual features from exceeding the LLM's maximum sequence length, we first set a budget $M$[5] for the total number of sub-images. We allocate this budget proportionally to each input image based on their original sizes. For each image **i** with dimensions $h_i \times w_i$, we calculate the initial number of sub-images $S_i$ as:

$$S_i = \left\lfloor \frac{h_i}{v} \right\rfloor \times \left\lfloor \frac{w_i}{v} \right\rfloor, \tag{1}$$

where $v$ is the resolution of visual encoder (e.g., $v = 364$ pixels). If the total number of patches satisfies $\sum_i S_i \leq M$, we proceed with these sub-image counts. Otherwise, we scale down these counts proportionally using a scaling factor $\alpha = \frac{M}{\sum_i S_i}$, resulting in adjusted sub-image counts:

$$S_i' = \lfloor \alpha S_i \rfloor. \tag{2}$$

**Image Partitioning**: For each image, we perform a grid search over possible number of rows $r$ and columns $c$ (where $1 \leq r, c \leq S_i'$ and $r \times c \leq S_i'$) to find the optimal cropping configuration that maximizes the effective resolution within the allocated sub-images (Li et al., 2024a). This configuration results in the original image being padded and resized to a target resolution of $(h_i' = r \times v, w_i' = c \times v)$. We then divide the image into $r \times c$ sub-images of size $(v \times v)$. Additionally, the original image is directly resized to $(v \times v)$, which provides a global view of the visual content.

**Image Encoding**: Most vision encoders transform an image into a sequence of visual features $\mathbf{v} \in \mathbb{R}^{L \times d}$, where $L$ represents the sequence length and $d$ denotes the feature dimension. Typically, $L$ is in the hundreds, *e.g.,* the `SigLIP` encoder yields a visual feature sequence in the shape of $L = 676$ and $d = 1152$ for the input image. Given that most LLMs have a sequence length of only 8K tokens, this implies that without any text input, the model can encode at most 12 images, which severely limits the image allocation budget. To mitigate this issue, inspired by the pixel shuffling operation (Chen et al., 2024a; Laurençon et al., 2024), we apply a similar strategy to the visual features. Specifically, we concatenate $n$ adjacent visual features along the feature dimension,

---

[5]$M$ is a hyperparameter, and we provide experiments on varying different $M$ in Figure 3.

Table 2: A detailed comparison of the model training details between baseline models and LEOPARD, including image resolution, vision encoder, backbone LLM, number of parameters (Param.), pre-training (PT.) data size, and instruction tuning (IT.) data size of baselines. AnyRes denotes the resolution selecting method proposed by Liu et al. (2024a) and Adapt HR. represents the proposed adaptive high-resolution multi-image encoding strategy.

| Models | Visual Encoder | Resolution | Backbone LLM | Param. | PT. | IT. |
|---|---|---|---|---|---|---|
| Otter-9B (Li et al., 2023) | CLIP ViT-L | $224^2$ | LLaMA-7B | 9B | 30M | 5.1M |
| Emu2-Chat (Sun et al., 2023) | EVA-02-CLIP | $448^2$ | LLaMA-33B | 37B | - | 160M |
| MM1-7B-Chat (McKinzie et al., 2024) | CLIP ViT-H | $378^2$ | - | 7B | - | 1.5M |
| VILA1.5-8B (Lin et al., 2023) | SigLIP | $384^2$ | LLaMA3-8B | 8B | 50M | 1M |
| mPlug-DocOwl-1.5 (Hu et al., 2024a) | CLIP ViT-L | $448^2$ (x9 crops) | LLaMA-7B | 8B | 4M | 1M |
| Idefics2-8B (Laurençon et al., 2024b) | SigLIP | $980^2$ | Mistral-7B | 8B | 350M | 20M |
| LLaVA-NeXT-Inter (Li et al., 2024c) | SigLIP | AnyRes | Qwen1.5-7B | 7B | 1.3M | 1.2M |
| Mantis-LLaVA (Jiang et al., 2024) | SigLIP | $384^2$ | LLaMA3-8B | 8B | 0.5M | 1M |
| Mantis-Idefics2 (Jiang et al., 2024) | SigLIP | $980^2$ | Mistral-7B | 8B | 350M | 1M |
| LEOPARD-LLaVA (Ours) | SigLIP | Adapt HR. | LLaMA3.1-8B | 8B | 0.5M | 1.2M |
| LEOPARD-Idefics2 (Ours) | SigLIP | $980^2$ | Mistral-7B | 8B | 350M | 1.2M |

effectively reducing the sequence length by a factor of $n$. This results in a compressed visual feature sequence $\mathbf{v}' \in \mathbb{R}^{\frac{L}{n} \times nd}$. By decreasing the sequence length in this way, we are able to accommodate more images within the sequence length constraints of the LLM. To incorporate visual features into the LLM, we first project the encoded visual feature sequences into the textual input embedding space using a vision-language connector. Since the partitioned images yield feature sequences of variable length, we introduce special tokens into the textual input to demarcate the image features to help the model distinguish visual features. Specifically, the sequence for the $i$-th image is formatted as: {Image $i$:  <Visual Feature Sequence> < /Img>}, where  and < /Img> are special tokens. An illustrative example of this sequence formatting is provided in Figure 2.

# 4 EXPERIMENT

## 4.1 IMPLEMENTATION DETAILS

**Model Architecture.** We train our models on two base architectures: LLaVA (Liu et al., 2023a) and Idefics2 (Laurençon et al., 2024b). For LEOPARD-LLaVA, we use SigLIP-SO-400M (Zhai et al., 2023) with $364 \times 364$ image resolutions as the visual encoder since it supports larger resolution than the commonly used $224 \times 224$ resolution CLIP visual encoder (Radford et al., 2021). Each image is encoded into a sequence of $26 \times 26 = 676$ visual features under a patch size of $14$. With the visual feature pixel shuffling strategy, each image is further processed into a sequence of 169 visual features. We limit the maximum number of images ($M$) in each sample to 50, which produces up to $8,450$ visual features in total. Following Liu et al. (2023a), we adopt a two-layer MLPs as the visual-language connector. We use LLaMA-3.1 (Meta et al., 2024) as the LM.

For LEOPARD-Idefics2, we follow the architecture of Idefics2-8B which uses SigLIP-SO-400M as the visual encoder but increases its image resolution to $980 \times 980$ to make the text legible. The features outputted by the visual encoder are compressed with a feature resampler into $64$ tokens per image. Idefics2-8B adopts the Mistral-7B (Jiang et al., 2023) as the LM.

**Training Details.** When training LEOPARD-LLaVA, we first train the visual-language connector using LLaVA's 558K multimodal pre-training dataset. Subsequently, we fine-tune the model (with both the connector and the LM unfrozen) using our LEOPARD-INSTRUCT data. As for LEOPARD-Idefics2, it is pre-trained on a dataset comprised of over 350M multimodal samples. Given the computational challenges of reproducing such extensive pre-training, and to ensure a fair comparison with baselines that utilize the pre-trained Idefics2 checkpoint, we directly adopt Idefics2' visual feature resampler and fine-tune the model on the LEOPARD-INSTRUCT dataset.

We train both LEOPARD-LLaVA and LEOPARD-Idefics2 on 64 A100-40G GPUs with a global batch size of 128. We use the AdamW optimizer with $\beta_1 = 0.9$, $\beta_2 = 0.999$. Following Jiang et al. (2024), we use a learning rate of $1 \times 10^{-5}$ for LEOPARD-LLaVA and $5 \times 10^{-6}$ for LEOPARD-

Table 3: Experiment results of baseline models and LEOPARD on 8 benchmarks of text-rich images. We use abbreviated benchmark names due to space limits. $MVQA^D$: Multi-page DocVQA, MCQA: MultiChartQA, MH: MultiHiertt, $VQA^T$: TextVQA, $VQA^D$: DocVQA, VWB: VisualWebBench. Following (Tito et al., 2022), for $MVQA^D$, DUDE, and $VQA^D$, we use average normalized leven-shtein similarity (ANLS) as the evaluation metric. For others, accuracy (Acc.) is used as the metric, which measures whether the predicted answer matches exactly with any of the target answers.

| Models | Text-Rich Multi-Image | | | | | | Text-Rich Single-Image | | | |
|---|---|---|---|---|---|---|---|---|---|---|
| | $MVQA^D$ | DUDE | SlideVQA | MCQA | MH | Multi Avg. | $VQA^T$ | $VQA^D$ | VWB | Avg. |
| Otter-9B | 0.17 | 0.15 | 5.95 | 1.08 | 0.14 | 1.50 | 23.18 | 3.53 | 10.20 | 12.30 |
| Emu2-Chat | 17.58 | 13.79 | 0.60 | 2.40 | 0.72 | 7.02 | 66.60 | 5.44 | 18.17 | 30.07 |
| MM1-7B-Chat | - | - | - | - | - | - | 72.80 | - | - | - |
| VILA-LLaMA3-8B | 30.75 | 19.75 | 24.72 | 1.87 | 3.66 | 16.15 | 66.30 | 30.38 | 23.37 | 40.02 |
| mPlug-DocOwl-1.5 | 35.85 | 16.94 | 4.54 | 0.26 | 0.86 | 11.69 | 68.60 | **82.20** | **29.80** | 60.20 |
| Idefics2-8B | 46.67 | 23.06 | 25.14 | 2.59 | 9.89 | 21.47 | 70.40 | 67.30 | 23.76 | 53.82 |
| LLaVA-NeXT-Inter | 39.92 | 24.04 | 23.46 | 14.34 | 3.55 | 21.06 | 62.76 | 75.70 | 21.36 | 53.27 |
| Mantis-LLaVA | 31.89 | 17.73 | 16.81 | 9.72 | 3.46 | 15.92 | 59.20 | 39.02 | 17.88 | 38.70 |
| Mantis-Idefics2 | 51.61 | 27.74 | 24.02 | 12.97 | 5.48 | 24.36 | 63.50 | 54.03 | 22.47 | 46.67 |
| LEOPARD-LLaVA | 53.90 | 35.94 | 23.83 | 9.68 | **10.76** | 26.82 | 67.70 | 68.07 | 24.91 | 53.56 |
| LEOPARD-Idefics2 | **66.06** | **40.74** | **34.93** | **18.03** | 10.09 | **33.97** | **80.40** | 74.79 | 25.60 | **60.26** |

Idefics2 to protect its pretrian knowledge. We use a cosine learning rate scheduler with a linear learning rate warm-up for the first $3\%$ steps. All model variants are trained 1 epoch under the same hyperparameters. It takes around 120 GPU days to train LEOPARD under both settings.

## 4.2 BASELINE MODELS

We compare LEOPARD against a range of existing open-source MLLMs that support multi-image inputs. The baseline models included in our comparison are Otter-9B (Li et al., 2023), Emu2-Chat-34B (Sun et al., 2023), MM1-7B-Chat (McKinzie et al., 2024), Mantis (Jiang et al., 2024), VILA (Lin et al., 2023), Idefics2-8B (Laurençon et al., 2024b), and LLaVA-NeXT-Interleave (Li et al., 2024c).

Models that only support a single image input are excluded from our comparisons, except for mPlug-DocOwl-1.5 (Hu et al., 2024a), as it is primarily trained on visual document data and demonstrates strong capabilities on text-rich image tasks. Table 2 demonstrates a detailed comparison of the model training details of between baseline models and our proposed LEOPARD, which highlights their architecture, image resolution and training data differences.

Table 4: Experimental results on general domain benchmarks. We abbreviate the Image split of ScienceQA as $SQA^I$.

| Models | MIRB | MiBench | MMMU | MathVista | $SQA^I$ | Avg. |
|---|---|---|---|---|---|---|
| Otter-9B | 20.74 | 43.72 | 30.89 | 22.00 | 60.44 | 35.55 |
| Emu2-Chat | 36.02 | 58.93 | 34.10 | 30.40 | 65.69 | 45.03 |
| MM1-7B-Chat | - | - | 37.00 | 35.90 | 72.60 | - |
| VILA-LLaMA3-8B | 40.87 | 53.70 | 36.90 | 35.40 | 79.90 | 49.35 |
| mPlug-DocOwl-1.5 | 25.39 | 40.80 | 35.44 | 29.50 | 64.40 | 39.11 |
| Idefics2-8B | 33.02 | 46.39 | 42.90 | 45.00 | 89.04 | 51.27 |
| LLaVA-NeXT-Inter | **44.38** | **74.52** | 38.44 | 32.10 | 72.63 | 52.41 |
| Mantis-LLaVA | 40.76 | 59.96 | 40.10 | 34.40 | 74.90 | 50.02 |
| Mantis-Idefics2 | 41.80 | 56.80 | 41.10 | 40.40 | 81.30 | 52.28 |
| LEOPARD-LLaVA | 42.00 | 60.80 | **43.00** | **45.50** | 85.57 | 55.37 |
| LEOPARD-Idefics2 | 41.38 | 61.74 | 40.11 | 44.80 | **90.38** | **55.68** |

## 4.3 EVALUATING BENCHMARKS

We evaluated LEOPARD and baseline methods across three categories of vision-language tasks on (1) single text-rich image evaluation, (2) multiple text-rich images evaluation, and (3) general reasoning evaluation. Benchmarks for (1) include TextVQA (Singh et al., 2019b), DocVQA (Mathew et al., 2021), and VisualWebBench (Liu et al., 2024c). Benchmarks for (2) include Multi-page DocVQA (Tito et al., 2022), DUDE (Landeghem et al., 2023), SlideVQA (Tanaka et al., 2023), Multihiertt (Zhao et al., 2022), and MultiChartQA (Anonymous, 2024), which cover a diverse range of typical multi-image tasks, such as document understanding and slide question answering. Benchmarks for (3) include MMMU (Yue et al., 2024), MathVista (Lu et al., 2024), ScienceQA (Saikh

Table 5: Ablation studies on LEOPARD-LLaVA from four different perspectives: (1) evaluating the impact of Adaptive High-Resolution Encoding, (2) pre-training LLaVA by initializing with checkpoints from either LLaMA-3 or LLaMA-3.1 , and (3) examining the impact of using different data domains for instruction tuning, including doc , chart , and web .

| Ablation Settings | Text-Rich Multi-Image | | | | Text-Rich Single | | General | |
|---|---|---|---|---|---|---|---|---|
| | MVQA$^D$ | DUDE | SlidesVQA | **Multi Avg.** | TextVQA | DocVQA | MMMU | MathVista |
| (⋆) *Our Best Setting (as in Table 3):* LLaMA-3.1 + Adaptive + ▢ ▢ ▢ | | | | | | | | |
| LEOPARD-LLaVA | 53.90 | 35.94 | 23.83 | **37.89** | 67.70 | 68.07 | 43.00 | 45.50 |
| (1) *Effect of Adaptive High-Resolution Encoding:* LLaMA-3.1 + ▢ ▢ ▢ | | | | | | | | |
| - w/o Adaptive | 40.44 | 26.16 | 20.93 | 29.17(**8.7**↓) | 60.18 | 44.69 | 41.00 | 42.40 |
| (2) *Effect of Backbone LLMs:* LLaMA-3 + Adaptive + ▢ ▢ ▢ | | | | | | | | |
| - with LLaMA-3.1 | 48.66 | 32.64 | 25.75 | 35.68(**2.2**↓) | 67.08 | 54.92 | 41.22 | 42.10 |
| (3) *Effect of Data Domains:* LLaMA-3.1 + Adaptive | | | | | | | | |
| - with chart web | 43.79 | 29.50 | 23.10 | 32.13(**5.7**↓) | 66.78 | 56.60 | 40.67 | 44.80 |
| - with doc web | 54.33 | 35.65 | 18.73 | 36.23(**1.7**↓) | 66.86 | 50.78 | 41.89 | 39.60 |
| - with doc chart | 54.62 | 35.70 | 20.79 | 37.02(**0.9**↓) | 67.40 | 67.82 | 41.78 | 44.00 |

et al., 2022), MIRB (Zhao et al., 2024) and MiBench (Liu et al., 2024b), which evaluate MLLMs from different perspectives, including world knowledge, mathematics, and scientific reasoning *etc.*

## 4.4 MAIN EXPERIMENTAL RESULTS

### *Question 1: How does LEOPARD compare to state-of-the-art MLLMs on vision-language tasks?*

LEOPARD achieves outstanding performance on **text-rich, multi-image** benchmarks, as shown in Table 3. Notably, both LEOPARD-LLaVA and LEOPARD-Idefics2 significantly outperform all baselines. LEOPARD-Idefics2 becomes the strongest open-source MLLM in this area, achieving an average improvement of 9.61 points over the previous best performance.

In **single-image text-rich** scenarios, LEOPARD outperforms several recent strong models, including VILA and LLaVA-NeXT. LEOPARD even achieves slightly higher average scores than the state-of-the-art mPlug model, despite mPlug being trained on 4M single-image data while LEOPARD is tuned on <200K. This demonstrates that training on multi-image data from LEOPARD-INSTRUCT also benefits model performance on single-image tasks.

In addition, we evaluate LEOPARD on **general-domain** benchmarks which contain both multi-image and single-image instances. As shown in Table 4, LEOPARD outperforms other open-source MLLMs on these benchmarks. Remarkably, LEOPARD surpasses Mantis, its counterpart multi-image model trained on the same foundational architecture and a comparable volume of data. This performance demonstrates the high quality and diversity of the LEOPARD-INSTRUCT dataset, which effectively preserves our model's general image understanding capabilities.

### *Question 2: Is the one-million text-rich multi-image dataset effective for instruction tuning?*

Mantis-Idefics2 is trained on a combination of natural *multi-image data* and *text-rich single-image* data. However, LEOPARD-Idefics2 outperforms Mantis-Idefics2 by 12.8 points on text-rich multi-image benchmarks. This disparity indicates that developing strong multi-image text-rich capabilities through cross-domain transfer, such as with Mantis data, presents significant challenges. This finding underscores the importance of optimizing LEOPARD using high-quality, diverse, and well-curated multi-image text-rich datasets that are specifically tailored for complex multi-image scenarios.

Furthermore, LEOPARD-Idefics2 surpasses its base model, Idefics2, by 6.4 points across three single-image text-rich benchmarks, though Idefics2 is trained on over 20M instruction data that includes text-rich tasks like DocVQA and TextVQA. This highlights that the LEOPARD-INSTRUCT provides unique advantages to MLLMs that are not adequately addressed by existing datasets.

### *Question 3: Does Adaptive high-resolution multi-image encoding improve MLLM performance?*

To assess the effectiveness of the proposed adaptive high-resolution multi-image encoding, we compared LEOPARD with a variant that excludes this feature (*i.e.*, *w/o* Adaptive in Table 5). We

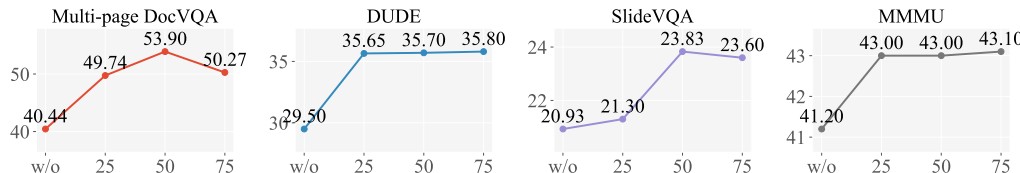

Figure 3: Impact of the sub-image budget $M$ on the resulting model across four benchmarks. *w/o* means original images are not partitioned into sub-images.

notice a significant performance decline across all text-rich benchmarks, particularly on document-related benchmarks like DocVQA (-23.4), Multi-page DocVQA (-13.5), and DUDE (-9.8). This observation supports our hypothesis that high-resolution image encoding is especially beneficial for text-rich images, particularly those with dense text content such as document pages.

### 4.5 MORE ANALYSIS

*Question 4: How does data from different domains contribute to instruction tuning?*

LEOPARD-INSTRUCT mainly cover three main domains, *i.e.,* documents & slides ( doc ), tables & charts ( chart ), and websites ( web ). To assess the impact of data from different domains, we conduct ablation studies on three variants of LEOPARD, with the results presented in Table 5 Removing any part of the training data results in performance degradation. The most significant drop occurs when we exclude document data while removing web data leads to a slight decrease. However, the mixed-domain datasets, such as LLaVAR and mPlugDocReason, also contain data in these domains which are challenging to isolate and ablate. This may contribute to the relatively preserved performance even after the ablation of certain data sources.

*Question 5: What is the influence of different image budgets in adaptive multi-image encoding?*

In our adaptive multi-image encoding module, we define a budget $M$ for the maximum number of sub-images that the model can process. To evaluate the impact of such image partitioning, we train LEOPARD using different values of $M$: 25, 50, 75, as well as a baseline setting where no image partitioning is applied and the number of sub-images equals the number of original images. According to the results plotted in Figure 3, model performance peaks or plateaus when $M$ is set around 50. Thus, we adopt 50 as the default value for training LEOPARD. These results show that increasing image numbers does not consistently improve performance, as input sequences can become excessively long and even exceed the model's sequence length limit.

*Question 6: How does the backbone language model affect the performance?*

To ensure a fair comparison with multi-image competitor models, Mantis-LLaVA and VILA1.5, we also evaluate a variant of LEOPARD using  LLaMA-3  instead of  LLaMA-3.1 , aligning its backbone language model architecture with these two baselines. According to Table 5, this substitution results in only a slight drop in average performance on text-rich multi-image tasks (2.2↓). Nevertheless, comparing with results in Table 3, LEOPARD-LLaMA-3 still substantially outperforms both baselines in all tasks, such as Multi-page DocVQA (+16.8 over Mantis and +17.9 over VILA) and DUDE (+14.9 over Mantis and +12.9 over VILA). These results indicate that LEOPARD's superior performance is not simply a result of the upgraded backbone large language models.

## 5 CONCLUSION

In this paper, we introduce LEOPARD, a novel MLLM specifically designed for text-rich, multi-image tasks. LEOPARD is equipped with two key innovations: (1) LEOPARD-INSTRUCT, a large-scale instruction-tuning dataset that encompasses a wide range of text-rich, multi-image instructions, and (2) an adaptive image encoding module capable of processing multiple high-resolution images efficiently. Our experimental results across diverse benchmarks highlight LEOPARD's superior performance compared to existing open-source MLLMs, particularly in text-rich multi-image scenarios. Further analysis and ablation studies underscore the effectiveness of both the collected dataset and adaptive encoding strategy, solidifying LEOPARD's contribution to multimodal research.

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

# A APPENDIX

## A.1 LEOPARD-INSTRUCT

To train LEOPARD, we created a large instruction-tuning dataset, LEOPARD-INSTRUCT, with **925K** instances, including **739K** designed for text-rich, multi-image scenarios. Despite surveying existing datasets, we found only **154K** suitable text-rich, multi-image samples – insufficient for effective instruction tuning, which is far from sufficient for effective instruction tuning, as shown in prior MLLM studies (Jiang et al., 2024; Laurençon et al., 2024b; Li et al., 2024c). To overcome this limitation, we developed several data collection pipelines to collect high-quality text-rich, multi-image data, resulting in additional **585K** instances.

Table 6 provides a detailed breakdown of the composition of the LEOPARD-INSTRUCT dataset. This table includes the name, domain, and sample size of sub-datasets. Additionally, it specifies how we construct multi-image samples, the number of images per sample, and the presence of rationales.

Table 6: Details of the constructed LEOPARD-INSTRUCT dataset. **Images** denotes the image number of one sample in each dataset.

| Dataset | Domain | Multi-image | Images | Rationales | #Samples (K) |
|---|---|---|---|---|---|
| ArxivQA (Li et al., 2024d) | Doc | Reformed | 1-3 | Existing | 81 |
| DUDE (Landeghem et al., 2023) | Doc | Public | 1-50 | Augmented | 23 |
| MP-DocVQA (Tito et al., 2022) | Doc | Public | 1-20 | Augmented | 36 |
| DocVQA (Mathew et al., 2021) | Doc | No | 1 | None | 39 |
| TAT-DQA (Zhu et al., 2022) | Doc | Reformed | 2-5 | Augmented | 13 |
| SlidesGeneration (Sefid et al., 2021) | Slides | Repurposed | 1-20 | Augmented | 3 |
| SlidesVQA (Tanaka et al., 2023) | Slides | Public | 20 | Augmented | 10 |
| Slideshare | Slides | Collected | 2-8 | Augmented | 3 |
| Multihiertt (Zhao et al., 2022) | Table | Public | 3-7 | Existing/Augmented | 15 |
| MultiTabQA (Pal et al., 2023) | Table | Public | 1-2 | Augmented | 6 |
| TableGPT (Li et al., 2024e) | Table | Split | 2 | Existing | 4 |
| TabMWP (Lu et al., 2023) | Table | No | 1 | Existing | 23 |
| ChartGemma (Masry et al., 2024) | Chart | Reformed | 1-4 | Existing | 65 |
| DVQA (Kafle et al., 2018) | Chart | Reformed | 1-3 | None | 200 |
| FigureQA (Kahou et al., 2018) | Chart | Reformed | 1-2 | None | 36 |
| ChartQA (Masry et al., 2022) | Chart | Reformed | 2 | Augmented | 32 |
| Pew_MultiChart | Chart | Collected | 2 | Augmented | 20 |
| Mind2Web (Deng et al., 2023) | Web | Split | 1-5 | None | 7 |
| WebsiteScreenshots (Aydos, 2020) | Web | No | 1 | Augmented | 2 |
| Omniact (Kapoor et al., 2024) | Web | No | 1 | None | 1 |
| RICO (Hsiao et al., 2024) | Web | Reformed | 1-4 | None | 25 |
| WebVision (Li et al., 2017) | Web | No | 1 | Existing | 1 |
| WebUI (Wu et al., 2023a) | Web | No | 1 | None | 19 |
| LLaVAR (Zhang et al., 2023) | Mix | No | 1 | Existing | 15 |
| MathV360k (Shi et al., 2024) | Mix | No | 1 | None | 38 |
| Monkey (Li et al., 2024f) | Mix | Reformed | 1-3 | None | 92 |
| MPlugDocReason (Hu et al., 2024a) | Mix | No | 1 | Existing | 25 |
| IconQA (Lu et al., 2021) | Other | Public | 1-6 | Augmented | 64 |
| InfographicVQA (Mathew et al., 2022) | Other | No | 1 | Augmented | 23 |
| MapQA (Chang et al., 2022) | Other | Reformed | 1-2 | None | 4 |
| **Total** | - | - | - | - | 925 |

We draw a chart to illustrate the data composition of LEOPARD-INSTRUCT dataset 4.

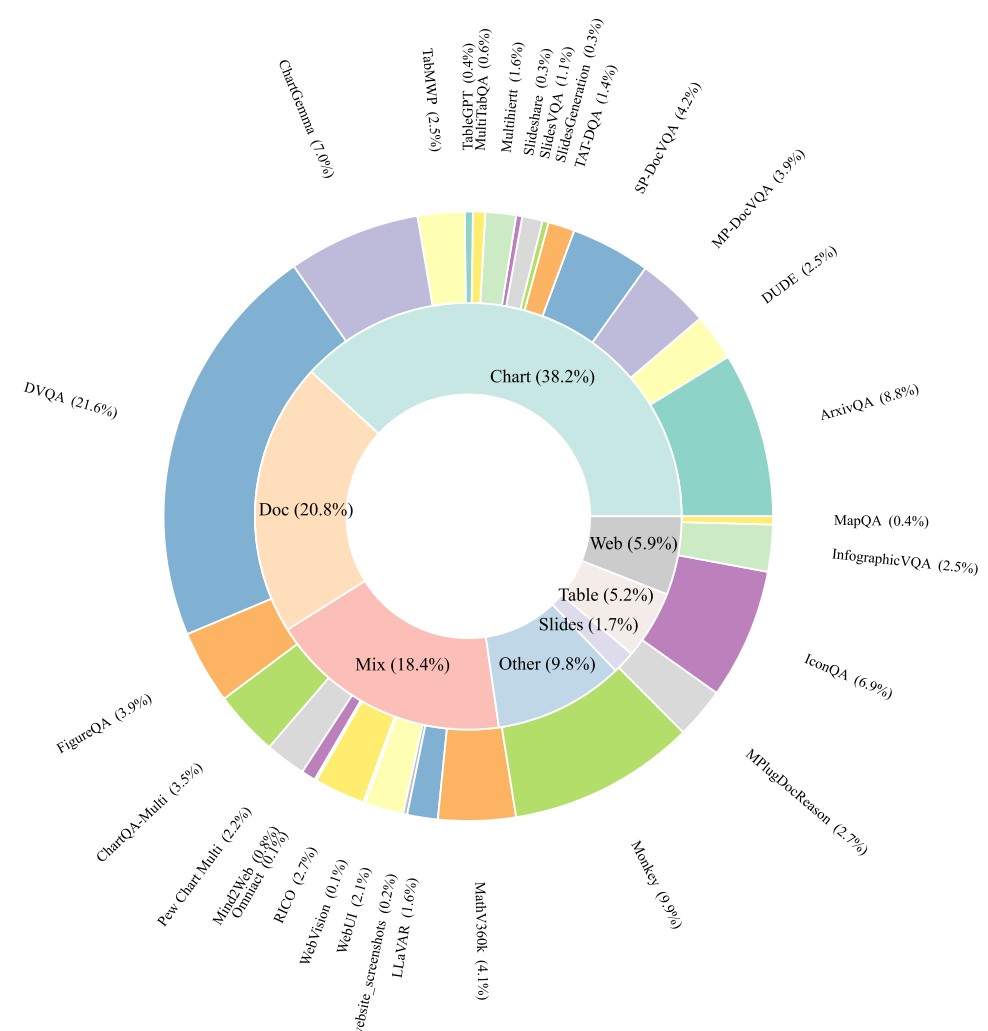

Figure 4: An illustration of the proportion of sub-datasets and domains in the proposed dataset.

## A.2 PROMPTS

We specify the prompt used during the data construction process as follows:

---

**Slides Q-A Generation Prompt**

You are given a set of images from a slides. Please generate 10 meaningful and distinct questions about the content of the slides.

You are supposed to generate the questions, the answers, and detailed explanations for the answers.
The questions should be clear, concise, and straightforward. The answers should be a few words or phrases.

You should ask questions about the details of the slides, including the tilte, the authors, and the figures and tables on the slides.

The output format should be in JSON format, with the following structure:
[{"Question_0":"...","Answer_0":"...","Rationale_0":"..."},
{"Question_1":"...","Answer_1":"...","Rationale_1":"..."}, ...]

---

Figure 5: The prompt used for generating Q-A pairs with rationales for slide decks data.

**Webpage Q-A Generation Prompt**

You are given a screenshot of a website. Please generate 10 meaningful and distinct questions about the screenshot. You should pay attention to the textual content, the layout, and the elements on the web screenshot.

You are supposed to generate the questions, the answers, and detailed explanations for the answers. The questions should be clear, concise, and straightforward. The answers should be a few words or phrases.

You should ask questions about the webpage description, the elements on the webpage, and the uses of buttons on the webpage.

The output format should be in JSON format, with the following structure:
[{"Question_0":"...","Answer_0":"...","Rationale_0":"..."},
{"Question_1":"...","Answer_1":"...","Rationale_1":"..."}, ...]

Figure 6: The prompt used for generating Q-A pairs with rationales for webpage data.

**Rationale Augmentation Prompt**

You are an expert in multi-page visual questions.
Based on the following question and answer, please generate a rationale that derives the answer.
### Question: {question}
### Answer: {answer}
### Rationale:

Figure 7: We use this prompt for the generation of chain-of-thought rationales given original question, answer, and images.

### A.3 DETAILS OF TABLE RENDERING

To convert the textual table dataset into a multimodal dataset, the JSON or DataFrame format data is transformed into tabular images using Python. We utilize three Python packages, *i.e.,* dataframe_image[6], pandas[7], and matplotlib[8] with various styling to enhance the diversity of the rendered images. To ensure the clarity and legibility of the plotted images, the original data is filtered by excluding any tables that contain more than 20 rows. This threshold was set to maintain the recognizability of the resulting images.

### A.4 QUALITATIVE RESULTS

We show two examples to give an illustrative demonstration of the model's performance. As can be seen from Figure 8, LEOPARD can not only capture detailed data in multiple tables precisely but also perform cross-table calculations, therefore it can answer the complex question correctly. Another example is demonstrated in Figure 9. LEOPARD can accurately perceive the prominent information under a high-resolution four-page document, demonstration effective text-rich abilities under multi-image scenarios.

---

[6] https://github.com/dexplo/dataframe_image.
[7] https://pandas.pydata.org/.
[8] https://matplotlib.org/.

**Image 1**

|  | For the years ended December 31, | For the years ended December 31,_1 | For the years ended December 31,_2 |
|---|---|---|---|
| 0 | 2013 | 2012 | 2011 |
| 1 Balance, beginning of period | $325 | $434 | $459 |
| 2 Sales inducements deferred | — | 7 | 20 |
| 3 Amortization — Unlock charge [1] | -72 | -82 | -28 |
| 4 Amortization charged to income | -33 | -34 | -17 |
| 5 Amortization charged to business dispositions [2] | -71 | — | — |
| 6 Balance, end of period | $149 | $325 | $434 |

**Image 2**

|  | 2014 | $200 |
|---|---|---|
| 0 | 2015 | 456 |
| 1 | 2016 | 275 |
| 2 | 2017 | 711 |
| 3 | 2018 | 320 |
| 4 | Thereafter | 4,438 |

**Image 3**

|  | Non-vested Units | Restricted Units (in thousands) | Weighted-Average Grant-Date Fair Value |
|---|---|---|---|
| 0 | Non-vested at beginning of year | 309 | 25.08 |
| 1 | Granted | — | — |
| 2 | Vested | -306 | 25.04 |
| 3 | Forfeited | -3 | 28.99 |
| 4 | Non-vested at end of year | — | $— |

**Image 4**

|  | December 31, 2013 | December 31, 2013_1 | December 31, 2012 | December 31, 2012_1 |
|---|---|---|---|---|
| 0 | Amount | Percent | Amount | Percent |
| 1 Fixed maturities, available-for-sale ("AFS"), at fair value | $62,357 | 79.2% | $85,922 | 81.6% |
| 2 Fixed maturities, at fair value using the fair value option ("FVO") | 844 | 1.1% | 1,087 | 1.0% |
| 3 Equity securities, AFS, at fair value | 868 | 1.1% | 890 | 0.8% |
| 4 Mortgage loans | 5,598 | 7.1% | 6,711 | 6.4% |
| 5 Policy loans, at outstanding balance | 1,420 | 1.8% | 1,997 | 1.9% |
| 6 Limited partnerships and other alternative investments | 3,040 | 3.9% | 3,015 | 2.9% |
| 7 Other investments [1] | 521 | 0.7% | 1,114 | 1.1% |
| 8 Short-term investments | 4,008 | 5.1% | 4,581 | 4.3% |
| 9 Total investments excluding equity securities, trading | 78,656 | 100% | 105,317 | 100% |
| 10 Equity securities, trading, at fair value [2] | 19,745 | | 28,933 | |
| 11 Total investments | $98,401 | | $134,250 | |

**Question**: What is the average value of Fixed maturities, available-for-sale ("AFS"), at fair value for Amount and Balance, beginning of period in 2013 ?

**LEOPARD**: To find the average value of Fixed maturities, available-for-sale ("AFS"), at fair value and Balance, beginning of period in 2013, we need to follow these steps:

1. Identify the values from the images:
   - Fixed maturities, available-for-sale ("AFS"), at fair value for Amount in 2013: $62,357 (from the fourth image).
   - Balance, beginning of period in 2013: $325 (from the first image).

2. Add these two values together:
   $[62,357 + 325 = 62,682]$

3. Divide the sum by 2 to find the average:
   $[\frac{62,682}{2} = 31,341]$

Thus, the average value of Fixed maturities, available-for-sale ("AFS"), at fair value and Balance, beginning of period in 2013 is $31,341.

Answer: 31341.0.

Figure 8: An example of multi-table reasoning of LEOPARD.

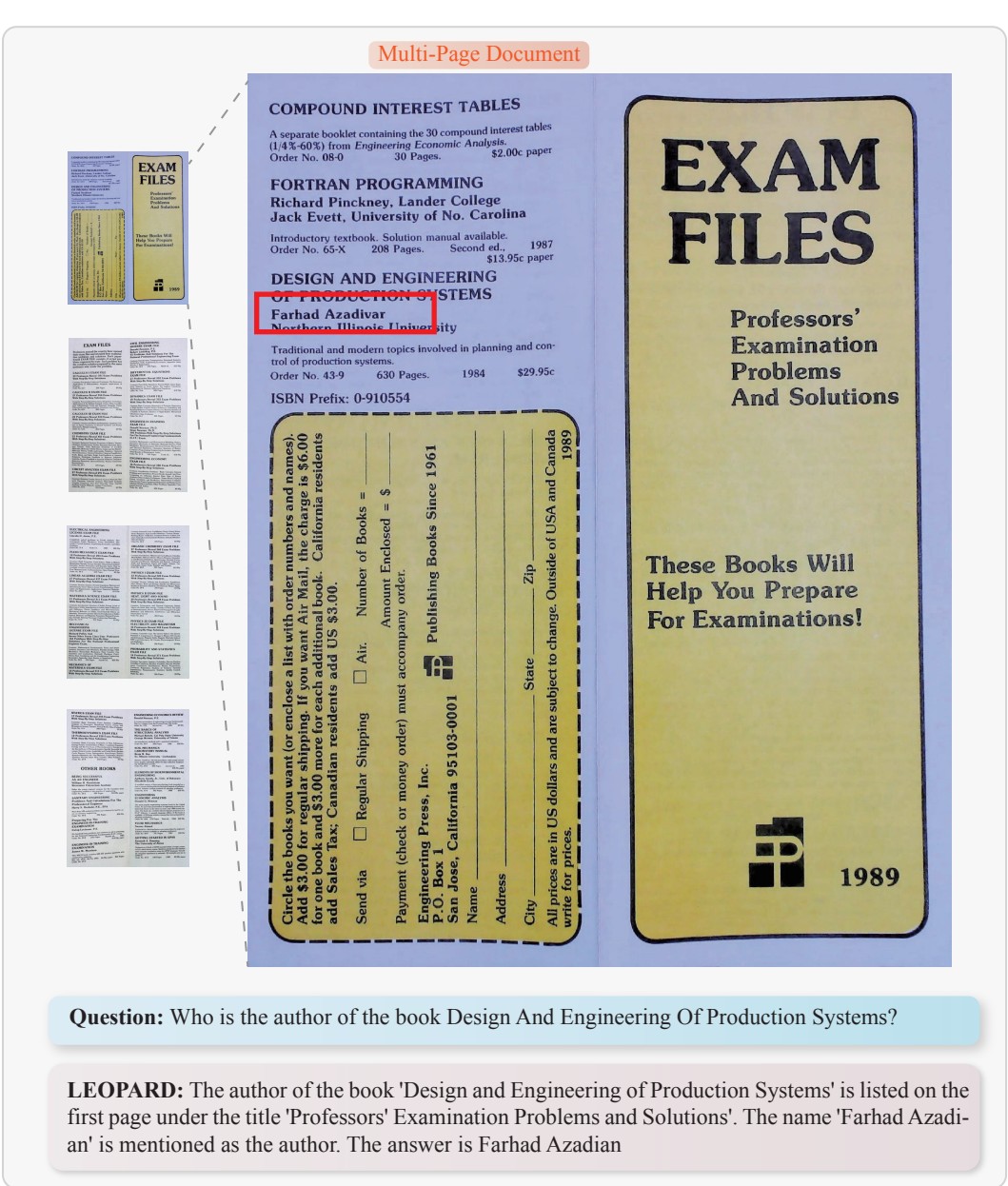

Figure 9: An example of multi-page document question answering of LEOPARD.

