# OpenReview forum: "Leopard: A Vision Language Model For Text-Rich Multi-Image Tasks"
_ICLR.cc/2025/Conference — ICLR 2025 Conference Withdrawn Submission_

### Official Review · Reviewer_mHvD · 2024-10-28

**Soundness:** 3
**Presentation:** 3
**Contribution:** 2
**Rating:** 5
**Confidence:** 4

**Summary:**

The paper introduces LEOPARD, an MLLM designed for text-rich multi-image tasks.
It addresses two key challenges: the lack of high-quality datasets for tuning models on such tasks and the difficulty in balancing image resolution with sequence length limitations.
The authors create LEOPARD-INSTRUCT, with ~1M data instances, tailored for text-rich, multi-image scenarios, and an adaptive high-resolution multi-image encoding module to optimize visual sequence length.
Experiments show LEOPARD outperforms existing MLLMs on text-rich multi-image benchmarks and maintains competitiveness on single-image tasks and general domain evaluations.

**Strengths:**

This paper has two major contributions:
1. LEOPARD-INSTRUCT: The paper creates a large-scale, high-quality multimodal instruction-tuning dataset specifically curated for text-rich, multi-image scenarios. This dataset addresses the scarcity of such resources and provides a robust foundation for training and improving MLLMs on complex, real-world tasks involving multiple interconnected images.
2. Adaptive High-Resolution Multi-Image Encoding: The paper develops an adaptive high-resolution multi-image encoding module that optimizes the allocation of visual sequence length based on input images' aspect ratios and resolutions. It allows LEOPARD to handle high-resolution images without compromising detail or clarity.

**Weaknesses:**

The paper's weaknesses include:
1. Lack of detailed quality control for data generated by GPT-4o, which could impact model performance
2. Limited comparison with the latest visual encoding strategies like Qwen2-VL's dynamic resolution approach.

**Questions:**

1. Seems there's no quality control strategies in the GPT-4o based data generation. In L226, the authors said that the error rate of GPT-4o generated QA is less than 10%. My questions include:
1.1 How do those errors look like? Hallucination?
1.2. How will those errors affect the model performance?
1.3. How to design some approaches to reduce such errors?
2. Recently, Qwen2-VL has proposed to process visual inputs under dynamic visual resolution (no patching). It would be better if the comparison between adaptive high-res encoding (proposed in this paper) and Qwen2-VL visual encoding strategy can be presented.
3. Some methods are missing in comparisons, such as Qwen2-VL, InternLM-XComposer2.5, InternVL2.

---

### Official Review · Reviewer_ikt2 · 2024-10-28

**Soundness:** 3
**Presentation:** 3
**Contribution:** 2
**Rating:** 3
**Confidence:** 4

**Summary:**

In this paper, the authors propose LEOPARD, an MLLM specifically designed for text-rich, multi-image tasks. To realize this, the authors collected large-scale training data LEOPARD-INSTRUCT.

**Strengths:**

Please refer to Questions

**Weaknesses:**

Please refer to Questions

**Questions:**

## Strength
1. The paper is well-written and easy to follow
2. The proposed method works well on the targeted tasks.

## Weakness.
My main concern is the limited contribution and novelty of this paper.
1. The paper contains two parts, the MLLM LEOPARD and its training data LEOPARD-INSTRUCT. However,

    a) the MLLM is a simple combination of an off-the-shelf multi-image strategy and an off-the-shelf high-resolution strategy. I could hardly find any novel point about it.

    b) The training data is a combination of current datasets and newly collected data. I do not find any unique design in the data collection process. I believe it would be helpful for the research community, but it is not a sufficient contribution to be a top conference paper.

2. The experiment results are not solid. There are many new MLLMs that could handle both multi-images and high-resolution images. For example, the NVLM, InternVL-2, etc. The methods compared in the paper are somewhat out of date.

---

### Official Review · Reviewer_db5r · 2024-11-04

**Soundness:** 3
**Presentation:** 3
**Contribution:** 2
**Rating:** 5
**Confidence:** 3

**Summary:**

This paper proposes LEOPARD, a vision language model specifically designed for handling text-rich multi-image tasks. The main contributions are: (1) A large instruction-tuning dataset of about 1M samples focused on text-rich multi-image scenarios, and (2) an adaptive high-resolution multi-image encoding module that dynamically allocates visual sequence length based on image characteristics. The paper addresses two key challenges: the scarcity of high-quality instruction datasets for text-rich multi-image tasks, and the difficulty in balancing image resolution with sequence length constraints. The authors evaluate their model across 13 benchmarks, demonstrating superior performance on text-rich multi-image tasks while maintaining competitive results on single-image and general vision-language tasks.

Overall, this could be a good contribution to the field, with some clarifications needed on the methodology and experiments. Given these clarifications in an author response, I would be willing to increase my score.

**Strengths:**

- The authors deliver a specified text-rich multi image instruction tuning dataset which is highly comprehensive with coverage of diverse domains (documents, slides, charts, tables, webpages).

- The paper presentation is clear and informative enough to demonstrate their key ideas.

**Weaknesses:**

- The proposed methodology and model seems just a combination of some existing works like llava-interleave and pixel shuffling. The novelty is just not clear enough to me at least based on current manuscript. While the adaptive high-resolution encoding is presented as novel, the paper lacks clear theoretical justification for why it should perform better than fixed allocation strategies.
The relationship to and improvements upon prior work need to be more explicitly discussed to establish the technical novelty beyond combining known methods.

- The experiments try to evaluate the contribution of  both the fine-tuning dataset and adaptive encoding method but the ablation studies presented are insufficient to validate the key claims. More comprehensive experiments are needed comparing different feature allocation strategies, various compression ratios, and alternative approaches to handling multiple high-resolution images. The contribution of the dataset versus the methodology is not clearly separated in the results.

**Questions:**

- What is the key innovation in the adaptive encoding? It seems that the hyperparameter to be adaptively chosen is M, so does that means it simply figure out a sub-image size that can fit in the sequence length of various MLLMs.

- How is such approach compared to the current token reduction approach? Is that fair comparison if baselines can only take part of the input tokens restricted by the sequence length when your approach can successfully feed all the tokens forward?

---

### Note · Authors · 2024-11-15

I have read and agree with the venue's withdrawal policy on behalf of myself and my co-authors.